# COVID-19 Vaccine Hesitancy and Personality Traits; Results from a Large National Cross-Sectional Survey in Qatar

**DOI:** 10.3390/vaccines11010189

**Published:** 2023-01-16

**Authors:** Shuja Reagu, Roland M. Jones, Majid Alabdulla

**Affiliations:** 1Hamad Medical Corporation, Doha P.O. Box 24144, Qatar; 2Weill Cornell Medicine—Qatar, Doha P.O. Box 24144, Qatar; 3Centre for Addiction and Mental Health, 1001 Queen Street West, Unit 3.4, Toronto, ON M6J 1H4, Canada; 4College of Medicine, Qatar University, Doha P.O. Box 2713, Qatar

**Keywords:** COVID-19, vaccine hesitancy, personality traits, Qatar

## Abstract

Attitudes to vaccination arise from a complex interplay of personal and environmental factors. This has been true for the COVID-19 vaccination attitudes too and understanding personal factors would help design immunisation strategies that help in infectious disease control. The five-factor model of personality has been established as a valid construct in exploring individual attitudes and traits. This institutional review board approved study explores the relationship between these five domains of personality and attitudes to COVID-19 vaccination in Qatar which has a migrant majority population. A cross-sectional survey was conducted in Qatar using an online survey link containing validated tools to measure vaccine hesitancy and personality traits. People from diverse ethnic and sociodemographic backgrounds, amounting to 5340 individuals, completed the self-report survey. After controlling for social and demographic variables, individuals scoring significantly higher on Conscientiousness were more likely to refuse the COVID-19 vaccination, while those scoring significantly lower on Openness to experience and Neuroticism were also more likely to refuse COVID-19 vaccination. Both groups of individuals scoring significantly higher and lower on Conscientiousness and Neuroticism, respectively, were more likely to trust their own research than trust endorsement of the COVID-19 vaccine from their doctor or healthcare organisation. The study highlights the highly complex and sometimes contradictory relationship between vaccine hesitancy and personality traits and makes a case for understanding this relationship better in order to inform successful immunisation strategies.

## 1. Introduction

Vaccine hesitancy varies across the globe and even across population subgroups [1,2]. For instance, the vaccine acceptance patterns for COVID-19 did not uniformly follow easy access to vaccines and established healthcare systems across high income countries [3,4]. On the other hand, the psychological profiles of individuals who were vaccine hesitant were similar across sociodemographic variances [5].

The final individual decision to go for or refuse personal or child vaccination arises from a complex and dynamic interplay of environmental, societal, cultural, political and individual psychological and attitudinal factors [1,2,6].

Similar complex interplay of environmental and individual themes has been reported for the COVID-19 vaccine hesitancy across the globe [7,8,9].

This underlies the importance of individual cognitive and attitudinal factors to vaccination; attempts to understand why individuals with relatively easy access to vaccinations come to hold their positions have the potential to overcome a vaccine hesitancy that is not related to environmental factors. These factors can be as diverse as mistrust of science [10], belief in alternate medicine [11,12], lack of confidence in vaccines and pharmaceutical companies [8,13], fear of side effects of vaccines and vaccine safety [14,15], and belief in conspiracy theories [16,17] have been shown to significantly affect individual attitudes to vaccination. 

The exploration of these individual dispositions is important to fully understand vaccine hesitancy which in turn would help develop informed and targeted vaccination campaigns which have been shown to be effective [18].

The five-factor model of personality is a recognised and validated assessment of an individual’s personality traits and attitudes and envisions the individual’s personality across five fundamental dimensions, those of Extraversion, Agreeableness, Conscientiousness, Neuroticism, and Openness to Experience. It builds upon research that sets personality as a hierarchical organisation of personality traits across these five dimensions and extensive research has provided support for the comprehensiveness of the model and has supported its applicability across observers and cultures [19]. In fact, studies seeking relationships between the five factor personality traits and vaccine hesitancy for COVID-19 and seasonal flu have shown inconsistent associations [5,20,21,22,23,24,25].

The present study attempted to investigate the relationship between these personality domains and vaccine hesitancy in Qatar. The setting of Qatar makes for an interesting factor and somewhat unique to other studies where such setting could not be replicated. Qatar is a migrant majority state with over 90% of the residents originating from other countries and working in Qatar as economic migrants [26]. Qatar additionally made the COVID-19 vaccine available relatively early and to all its resident population free of cost [27]. Individuals in Qatar therefore come from countries where access to vaccines varies and they may have differing environmental factors. The sociodemographic factors underlying vaccine hesitancy have been reported by the authors in a published study and this unique setting will therefore offer a unique opportunity to evaluate a set of individuals from different sociocultural backgrounds and explore whether there are any common psychological factors that cut across cultures [14].

## 2. Methodology

### 2.1. Study Design

The authors conducted a convenience sampling, cross-sectional survey in Qatar between 15 October and the 15 November 2020 using an online survey available to all residents in Qatar. The link to the survey was advertised through online local newspapers, and across the social media platforms of the Hamad Medical Corporation, which is Qatar’s state funded (and by far the largest) healthcare provider for the country. The advertisements were accompanied by short videos in English and Arabic explaining the rationale and nature for the survey. The survey was available in both English and Arabic languages.

### 2.2. Participants

All 2.3 million adult residents of Qatar [26] were eligible for the study and were invited to participate in the survey.

### 2.3. Study Measures

A validated vaccine hesitancy measurement tool, The Vaccine Attitudes Examination Scale (VAX) [28], was used as part of a composite questionnaire to assess vaccine attitudes, awareness, and hesitancy among the study participants. This tool was translated into Arabic and validation of the translated version was carried out using the guideline published by Sousa and Rojjanasrirat [29]. This translation process involved using native speakers of both languages, bilingual proficient users of English and Arabic, and going through steps 1 to 6 of the guideline.

Vaccine hesitancy was measured by the question, “Will you take the COVID-19 vaccine when it is available?” Responses were given on a 5-point scale (Definitely/probably/unsure/probably not/definitely not). We divided the responses into three groups, (Probably/definitely accept, unsure, and probably/definitely reject) for analysis, and subsequently we created a binary variable for vaccine hesitators (those who would definitely or probably decline the vaccine as well as those who were unsure).

Personality was measured using the Ten Item Personality Inventory (TIPI) [30], which is a brief measure of the five-factor personality traits (also known as the “Big 5”). Although somewhat inferior to standard multi-item instruments, the instrument has demonstrated adequate confidence in terms of convergence with other five-factor measures of personality, and reliability [30]. Using this instrument, participants were asked to what extent various personality characteristics apply to them on a 7-pont scale from “Strongly agree” to “Strongly disagree”. Each of the “Big 5” personality traits (Extraversion, Neuroticism, Openness, Conscientiousness, and Agreeableness) were measured using two questionnaire items which each measured opposite dimensions of the trait: “extraverted, enthusiastic” and “reserved, quiet” for Extraversion; “sympathetic, warm” and “critical, quarrelsome” for Agreeableness; “dependable, self-disciplined” and “disorganised, careless” for Conscientiousness; and “anxious, easily upset” and “calm, emotionally stable” for Neuroticism. The scoring on the second scale of each paring was reversed, and a mean of the two scales was taken as the score for each primary personality trait. Each trait score therefore has a possible range of 1–7, with a neutral score (neither agree or disagree) centered at 4.

The survey also collected relevant demographic and contextual information of the participants including respondent type (health care worker or general public), age, nationality (Qatari or non-Qatari), highest level of education, occupation (salaried, self-employed or unemployed), marital status, and gender. In addition, the survey collected responses to questions as to whether respondents had received all childhood vaccinations (completed or not completed), whether they had received the influenza vaccine regularly in the last three years (annually, once or twice, or never), and what would make the individual more confident in accepting the vaccine (endorsement by their doctor or hospital/endorsement by a public figure/endorsement by the Ministry of Health/endorsement by the World Health Organisation/positive feedback from friends or family/reading scientific research of its effectiveness/other).

The selection of study tools (VAX) and the design of the composite questionnaire for the overall study was guided by the SAGE group recommendations in assessing vaccine hesitancy [6]. These are given in detail in the previous publication focusing on sociodemographic factors [14].

### 2.4. Ethical Approval

The study was granted ethical approval by the Medical Research Council of the Hamad Medical Corporation. (MRC approval-01-20-930).

### 2.5. Analysis

We analysed the data using descriptive statistics, one-way ANOVAs and multivariable logistic regression using Stata 14 [28]. We carried out one-way ANOVAs to compare the effect of personality trait on vaccine attitudes, and post hoc comparisons using the Sidak test. We then created a binary variable of vaccine hesitators (those who said they probably or definitely would not take the vaccine as well as those unsure) compared with those accepting (probably or definitely would take the vaccine) and carried out logistic regression to investigate the effect of variables on vaccine hesitation, controlling for available social and demographic variables.

We then analysed possible sources of influence that may make the person more likely to accept the vaccine. The options were divided into: endorsement by a doctor; endorsement by a public authority or by family; and carrying out own research. We carried one-way ANOVAs to compare the effect of personality trait on sources of influence. For comparison, we also investigated the relationship between the Big Five personality traits and acceptance of flu vaccine in the past 3 years.

## 3. Results

A total of 7859 individuals participated in the survey, however personality measures were available from only 5340 respondents. Our total sample size was thus 5340 for all analyses. There were no missing data on any measures in this sample. Sociodemographic details, vaccine hesitancy rates, and their interrelationship is detailed in Table 1.

The rates of vaccine hesitancy were around 20% of those surveyed and a further 20% were unsure about their decision to go for vaccination.

We carried out one-way ANOVAs to compare the effect of personality traits on vaccine attitudes (see Table 2). There was a significant effect of Extraversion on vaccine attitudes at the *p* < 0.001 level of significant (F(2,5337) = 44.7, *p* < 0.001). A post hoc comparison using the Sidak test indicated that the vaccine rejectors had higher levels of Extraversion than the undecided (0.23, *p* < 0.001) and acceptors (0.12, *p* = 0.025). Vaccine acceptors also had higher levels of Extraversion that the undecided (0.11, *p* = 0.043).

We found a significant effect of Openness on vaccine attitudes at the *p* < 0.001 level (F(2,5337) = 47.17, *p* < 0.001). Vaccine acceptors had significantly higher Openness than those who where unsure (0.36, *p* < 0.001) or rejectors (0.35, *p* < 0.001). We found a significant effect of Conscientiousness on vaccine attitudes (F(2,5292) = 20.96, *p* < 0.001); vaccine decliners had significantly higher Conscientiousness than the undecided (0.19, *p* < 0.001), and the acceptors (0.26, *p* < 0.001). We also found a significant effect of Neuroticism on vaccine attitudes (F(2,5337) = 44.7, *p* < 0.001); vaccine decliners had significantly lower Neuroticism than the undecided (−0.46, *p* < 0.001) and the acceptors (−0.40, *p* < 0.001). Finally, we saw a weak effect of Agreeableness on vaccine attitudes (F(2,5337) = 3.03, *p* = 0.049); however, post hoc testing did not reveal a significant differences between groups, though the greatest difference was between refusers and acceptors (0.10, *p* = 0.06).

We then investigated the relationship between vaccine hesitators and personality characteristics while controlling for social and demographic variables (see Table 3). We found that higher Conscientiousness was significantly associated with being a vaccine hesitator (odds ratio (OR) 1.13, 95% CI 1.07–1.20). We found that lower Openness was associated with vaccine hesitation (OR 0.81, 95% CI 0.77–0.85), as was lower Neuroticism (OR 0.87, 95% CI 0.83–0.91).

Regarding the question of what would make the person more likely to accept the vaccine, the options were divided into endorsement by a doctor, endorsement by a public authority or organisation, and own research (we excluded the “other” category from this analysis). We carried out one-way ANOVAs to compare the effect of personality trait on sources of influence (see Table 4). There was a significant effect of Extraversion on influence at the *p* = 0.002 level of significance (F(2,453) = 6.24). Post hoc comparison using the Sidak test indicated that those who reported that endorsement by a doctor would make them more likely to accept the vaccine had higher levels of extraversion than those more likely to be influenced by public figures authorities (1.61, *p* < 0.001).

We found a significant effect of Openness on attitudes to endorsement (F(2,4530) = 7.06, *p* < 0.001). Those preferring their own doctor’s opinion had lower Openness than those relying on public figures (−0.18, *p* = 0.001), or those preferring their own research (−0.11, *p* = 0.046).

We found Conscientiousness had a significant effect on source of influence (F(2,4493) = 10.92, *p* < 0.001). Those who preferred their own research had higher Conscientiousness than those who preferred endorsement by a doctor (0.12, *p* = 0.02), or by a public authority (0.18, *p* < 0.001). Finally, we found a significant effect of Neuroticism on attitudes to endorsement (F(2,4532) = 16.08, *p* < 0.00) those who preferred their own research had lower Neuroticism than those who preferred public figure endorsement (−0.36, *p* < 0.001), and those who preferred a doctor’s endorsement had lower Neuroticism than those who preferred public figures (0.14, *p* = 0.01).

For comparison, we investigated the association between personality and flu vaccination using one-way ANOVAs (see Table 5).

We found a significant effect of Openness on flu vaccine use (F(2,5447) = 6.62, *p* = 0.001). Those who had annual flu vaccines had higher Openness than those who had had it once or twice (0.15, *p* = 0.007) or had never had it (0.15, *p* = 0.002). Finally, we found Conscientiousness had a significant effect flu vaccine (F(2,292) = 8.07, *p* = 0.003). Those who had annual flu vaccines had lower Conscientiousness than those who had it once or twice (−0.14, *p* = 0.004) or never (−0.15, *p* < 0.001). 

## 4. Discussion

This is the first and the largest study of its kind in this region that reports on the relationship of personality traits to COVID-19 vaccine hesitancy. The main findings of this study are that COVID-19 vaccine hesitators reported significantly higher scores in the domains of Conscientiousness, but lower for Openness to experience and Neuroticism, when compared to vaccine acceptors even after controlling for social and demographic variables.

Additionally, significantly higher Conscientiousness scores were also reported by individuals who had been declining their annual flu vaccinations and preferred their own research to trusting their doctors or healthcare authorities. While individuals scoring significantly lower scores on Openness to experience had also declined annual flu vaccinations, they showed more trust in their doctor or healthcare provider over their own research.

The findings of higher Conscientiousness scores among vaccine hesitators in this study are in contrast to the findings were reported by some other researchers exploring COVID-19 vaccine hesitancy during the relatively earlier period of the pandemic, or exploring vaccine hesitancy for childhood vaccines before the pandemic. However, studies that were conducted relatively later are reporting results that show an inverse relationship between conscientiousness and vaccine hesitancy, which is in line with our findings [20,21,22]. For instance, Howard and colleagues [21] surmised that it could be that conscientious individuals believed their discipline with respect to healthy behaviours like social distancing could compensate for refusing vaccination. Additionally, several studies [20,22], in direct contradiction to previous studies, found high scores for Conscientiousness and lower Neuroticism associated with vaccine hesitancy.

We surmise that these results can be explained, in part, by the nature of the COVID-19 pandemic. This pandemic has been unprecedented in its impact and has been associated with fear arising from its rapid spread and the unknown aspects of the infection [31]. With technological advances, the speed of vaccine development has been equally unprecedented [32], which has fanned conspiracy beliefs [33]. The concerns around the safety of the vaccine and its long term impact were reported widely, including by the participants in this study [14,34]. We surmise that conscientious individuals may be overconfident in their ability to prevent the infection and to reach their own conclusions about vaccine safety [21,35]. This is borne out by the finding that scores for Conscientious were significantly higher for those who preferred to do their own research than believe messages from healthcare bodies.

Individuals with highest Neuroticism scores were unsure whether to accept or deny the COVID-19 vaccine. High Neuroticism scores have been associated with lack of confidence [35]; this may be reflected in individuals who could not commit to either to the fear of the infection itself or to negative beliefs about the vaccine and the fear of long term adverse effects. The fact that acceptors were significantly more likely to trust their doctor or healthcare body than their own research gives some credence to this surmise.

The trait of Extraversion has been associated with overconfidence, narcissism, and optimistic disposition [35]. As with Conscientiousness, individuals scoring significantly higher in this domain were more likely to be hesitators than acceptors of the COVID-19 vaccines, albeit not after controlling for other social and demographic variables.

Openness to experience has been associated with innovative and ultimately positive, health behaviours [36]. Openness to experience has also been the most strongly associated of the Big Five with liberal political orientations [37] and individuals with such attitudes have been more likely to be COVID-19 vaccine acceptors [38]. It is possibly what explains the significantly higher scores for Openness to experience in COVID-19 vaccine acceptors in this group mirrors what has been reported elsewhere in as disparate populations as those from the USA and China [39,40].

Overall, taking into consideration the results from this study and comparing them to what has been published elsewhere, it is clear that the relationship between personality traits and attitudes to vaccination is not a straightforward one. These relationships are complex, sometimes contradictory, and have the potential to evolve and change even within the same population groups. Additionally, it is becoming clear that it is important to explore this relationship, both within and outside the special circumstances of the COVID-19 pandemic, via well-designed studies and research questions that are informed by what we have learned so far. Future study designs should take into consideration the role and impact of social media on pro- or anti-vaccination attitudes as this plays an increasingly important role in influencing individual and popular opinions. It is only after we are confident in our understanding of this complex interplay that we can design well-informed vaccination campaigns.

## 5. Limitations

The study was cross-sectional, the tool was self-report, and the study population self-selecting, thereby limiting any comments on causality. Given the high proportion of non-Qatari and university educated respondents in the sample, it is not representative of the general population. Additionally, the study was conducted at a time when the vaccine had not yet been made available to the residents of Qatar and it is safe to surmise that attitudes to vaccination have changed since then.

## Figures and Tables

**Table 1 vaccines-11-00189-t001:** Sociodemographic and contextual data of the participants for COVID-19 vaccine hesitancy in Qatar.

		Frequency (%)
Respondent Type	Health care workers	1113 (20.84)
	General public	4227 (79.16)
Age Group	18 to 25	161 (3.01)
	26 to 35	1673 (31.33)
	36 to 45	1862 (34.87)
	46 to 55	841 (15.75)
	56 to 65	593 (11.10)
	Over 65	210 (3.93)
Nationality	Qatari	653 (12.3)
	Non-Qatari	4687 (87.7)
Educational Level	High school	432 (8.09)
	University	4263 (79.83)
	Trade/vocational/other	192 (3.60)
	Other	453 (8.48)
Occupation	Salaried	4185 (78.37)
	Self employed	273 (5.11)
	Unemployed	569 (10.66)
	Retired	313 (5.86)
Marital Status	Single	954 (17.87)
	Married	4386 (82.13)
Gender	Male	3061 (57.32)
	Female	2279 (42.68)
Childhood Vaccination Status	Completed	5054 (94.64)
	Not completed	286 (5.36)
How Often Received Flu Vaccine in Past 3 Years	Annually	1298 (24.3)
	Once or Twice	1394 (26.11)
	Never	2648 (49.6)
Will You Have the COVID-19 Vaccine When It Is Available?	Definitely	2309 (43.24)
	Probably	871 (16.31)
	Not Sure	1044 (19.10)
	Probably not	504 (9.44)
	Definitely not	612 (11.46)
What Would Make You More Confident in Receiving the Vaccine?	Endorsement by Doctor/Hospital	1258 (23.6)
	Endorsement by Public Figure/Organisation	1852 (34.7)
	Own reading/research	1458 (27.3)
	Other	772 (14.5)

**Table 2 vaccines-11-00189-t002:** Relationship between Big Five personality traits and likelihood of accepting COVID-19 vaccine.

	Accept		Unsure		Decline		F (2,5337)	*p*
	M	SD	M	SD	M	SD		
Extraversion	4.44	1.29	4.32	1.28	4.56	1.37	8.65	<0.001
Openness	5.12	1.26	4.76	1.30	4.77	1.42	47.17	<0.001
Conscientiousness	5.89	1.14	5.60	1.13	6.14	1.09	20.96 *	<0.001
Neuroticism	2.94	1.31	3.02	1.31	2.55	1.32	44.70	<0.001
Agreeableness	5.25	1.18	5.30	1.11	5.34	1.22	3.03	0.049

* F(2,5292). M = mean; SD = standard deviation.

**Table 3 vaccines-11-00189-t003:** Multivariable logistic regression model of the relationship between vaccine hesitators and Big-five personality traits and demographic characteristics.

	Odds Ratio	95% Confidence Interval	z	*p*
Age				
18–25	1.00			
26–35	0.984	0.68–1.42	−0.09	0.932
36–45	1.14	0.78–1.66	0.68	0.494
46–55	1.027	0.69–1.52	0.14	0.892
56–65	0.914	0.60–1.38	−0.43	0.668
>65	0.634	0.38–1.05	−1.78	0.075
Education				
High School	1.00			
University	0.657	0.44–0.97	−2.09	0.036
Trade/vocational	1.002	0.80–1.25	0.02	0.986
Other	0.773	0.58–1.04	−1.71	0.086
Occupation				
Salaried	1.00			
Self-employed	1.436	1.10–1.88	2.66	0.008
Unemployed	1.062	0.87–1.30	0.58	0.563
Retired	1.809	1.32–2.47	3.72	<0.001
Married				
No	1.00			
Yes	0.893	0.76–1.05	−1.34	0.181
Gender				
Male	1.00			
Female	2.37	2.07–2.70	12.89	<0.001
Nationality				
Non-Qatari	1.00			
Qatari	2.464	2.03–3.00	9.05	<0.001
Type of Encounter				
Healthcare Worker	1.00			
General public	1.02	0.87–1.18	0.22	0.833
Extraversion	1.00	0.95–1.03	−9.16	0.869
Conscientiousness	1.13	1.07–1.20	4.18	<0.001
Agreeableness	0.97	0.92–1.02	−1.24	0.216
Openness	0.81	0.77–0.85	−8.51	<0.001
Neuroticism	0.87	0.83–0.91	−5.66	<0.001

**Table 4 vaccines-11-00189-t004:** Relationship between influence of personal or professional endorsement of COVID-19 vaccination and Big Five personality traits.

	Doctor		Public Organisation/Figure		Own Research		F	*p*
	M	SD	M	SD	M	SD		
Extraversion	4.52	1.27	4.36	1.28	4.47	1.36	6.24	0.002
Openness	4.89	1.34	5.07	1.24	5.02	1.29	7.06	<0.001
Conscientiousness	5.94	1.16	5.88	1.15	6.06	1.05	10.92	<0.001
Neuroticism	2.88	1.34	3.02	1.31	2.76	1.30	16.08	<0.001
Agreeableness	5.34	1.18	5.24	1.14	5.26	1.18	2.57	0.076

(M = mean, SD = standard deviation).

**Table 5 vaccines-11-00189-t005:** Relationship between past Flu vaccination attitudes and Big Five personality traits.

	Annually		Once or Twice		Never		F	*p*
	M	SD	M	SD	M	SD		
Extraversion	4.47	1.30	4.45	1.31	4.43	1.31	0.13	0.882
Openness	5.09	1.25	4.94	1.32	4.94	1.36	6.62	0.001
Conscientiousness	5.85	1.17	5.99	1.11	5.99	1.12	8.07	<0.001
Neuroticism	2.84	1.24	2.95	1.32	2.86	1.35	2.74	0.065
Agreeableness	5.24	1.17	5.27	1.18	5.30	1.17	1.46	0.232

(M = mean, SD = standard deviation).

## Data Availability

Data is available on receipt of any reasonable request from the authors.

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
