# Peer review of "COVID-19 Vaccine Hesitancy and Personality Traits; Results from a Large National Cross-Sectional Survey in Qatar"

_vaccines, 2023, doi:10.3390/vaccines11010189_

Round 1

Reviewer 1 Report

This paper studies the relationship between personality traits and hesitancy in relation to the COVID-19 vaccine as measured by an electronically self-administered questionnaire in a cross-sectional study (population-based survey); 7821 individuals responded.

In my opinion the topic addressed by the paper is relevant and interesting but the study is exposed to numerous biases that call into question its validity.

INTRODUCTION

It would be interesting to explain in more detail the five-factor model of personality, especially the characteristics of the five fundamental dimensions: Extraversion, Agreeableness, Conscientiousness, Neuroticism, and Openness to Experience.

METHODOLOGY

In my opinion, the type of sampling explained does not ensure the representativeness of the study sample in relation to the general population and is excessively exposed to biases. The methodology does not explain the mechanisms used to mitigate these possible biases.

It should be clarified in which language the TIPI (Ten Item Personality Measure) was used; if an Arabic version was used, is it validated?

Measurement of the variables, although it refers to an already published article, a brief description of the questionnaire and its variables would be advisable.

ANALYSIS

Should better explain the statistical analysis and how variables have been selected for the regression models.  Some parts of the statistical analysis are described in the results section, should be corrected.

Describe all statistical methods, including those used to control for confounding factors.

Describe the methods used to examine subgroups and possible interactions.

Explain how missing data were treated.

RESULTS:

The high percentage of health care workers in the sample studied (Table 1) and the very high percentage of university students at the educational level are striking. This could be a source of selection bias to be taken into account but which the authors do not analyze or comment on. They should reflect on the composition of the sample obtained and its representativeness of the general population.

The explanation of the measurement of personality traits (lines 117 to 127) should be included in the methodology section.

DISCUSSION:

The discussion is difficult to follow

LIMITATIONS

In this section, the different sources of bias to which this paper is exposed and possible corrective measures adopted to minimize this possibility of error should be analyzed in more depth.

Reviewer 2 Report

I read with great interest the manuscript, which falls within the aim of this Journal. In my honest opinion, the topic is interesting enough to attract the readers’ attention. The overall quality of the presentation enhances the readability of the survey. While English is used generally correctly from a grammar and syntax point of view. The survey fully clarifies its position in the current literature on this domain 

Nevertheless, the authors should clarify some points and improve the discussion, as suggested below:

·        I have noticed a strange punctuation method for dots. For example, in lines 27, 48, 55,66,72 the dots are at the end of the sentence, after the parenthesis. But, for example, in lines 29,33,36, and 59 the dots are before the parenthesis. I think it would be better for the article to have the right way of writing.

·       Line 108 using Stata 12 (28)” -> please explain

·       At line 188 final part of the parenthesis is missing.

·       At line 90 is written, by Sousa et al” but in the parenthesis  (Sousa & Rojjanasrirat, 2011) ?

I would advise you to write, in brief, or a figure, about the steps of translation (Sousa & Rojjanasrirat, 2011), adaptation, and validation of the instrument.

The biggest concern is the sample size in the study. Could you please elaborate a bit about representativity?

The most important thing in the Discussion section is the need for mentioning the literature references from the introduction and evaluating and interpreting the results and this literature. As far as I can see there are a plethora of references in the Discussion section (Schaefer et al., 2004), (Jennings et al., 2021) that the authors have not mentioned before, in the introduction.

The future scope of this study can be added as well as the social impact can also be discussed in this paper.

Reviewer 3 Report

Dear authors,

your manuscript  is clear , easy to reading  , well structured  and takes home a message to help infectious diseases control

I think is appropriate for publication by Journal Vaccines. 

.

Author Response

Dear authors,

your manuscript  is clear , easy to reading  , well structured  and takes home a message to help infectious diseases control

I think is appropriate for publication by Journal Vaccines. 

Response: The authors are grateful to the reviewer for their valuable time and their kind comments and are glad that the reviewer found the paper ready for publication.

Round 2

Reviewer 1 Report

Dear Authors

Thank you for having taken into consideration the comments I have made as a reviewer.